# Computing nasalance with MFCCs and Convolutional Neural Networks

**Andrés Lozano**[1]*, **Enrique Nava**[1], **María Dolores García Méndez**[2], **Ignacio Moreno-Torres**[2]

1 Department of Communication Engineering, University of Málaga, Málaga, Spain, 2 Department of Spanish Philology, University of Málaga, Málaga, Spain

* ald@uma.es

**Data Availability Statement:** All autor-generated code script files can be found in public repository: https://doi.org/10.6084/m9.figshare.26762818.v1 https://doi.org/10.6084/m9.figshare.26762842.v1 https://doi.org/10.6084/m9.figshare.26762833.v1.

## Abstract

Nasalance is a valuable clinical biomarker for hypernasality. It is computed as the ratio of acoustic energy emitted through the nose to the total energy emitted through the mouth and nose (*eNasalance*). A new approach is proposed to compute nasalance using Convolutional Neural Networks (CNNs) trained with Mel-Frequency Cepstrum Coefficients (*mfccNasalance*). *mfccNasalance* is evaluated by examining its accuracy: 1) when the train and test data are from the same or from different dialects; 2) with test data that differs in dynamicity (e.g. rapidly produced diadochokinetic syllables versus short words); and 3) using multiple CNN configurations (i.e. kernel shape and use of 1 × 1 pointwise convolution). Dual-channel Nasometer speech data from healthy speakers from different dialects: Costa Rica, more(+) nasal, Spain and Chile, less(-) nasal, are recorded. The input to the CNN models were sequences of 39 MFCC vectors computed from 250 ms moving windows. The test data were recorded in Spain and included short words (-dynamic), sentences (+dynamic), and diadochokinetic syllables (+dynamic). The accuracy of a CNN model was defined as the Spearman correlation between the *mfccNasalance* for that model and the perceptual nasality scores of human experts. In the same-dialect condition, *mfccNasalance* was more accurate than *eNasalance* independently of the CNN configuration; using a 1 × 1 kernel resulted in increased accuracy for +dynamic utterances (p < .000), though not for -dynamic utterances. The kernel shape had a significant impact for -dynamic utterances (p < .000) exclusively. In the different-dialect condition, the scores were significantly less accurate than in the same-dialect condition, particularly for Costa Rica trained models. We conclude that *mfccNasalance* is a flexible and useful alternative to *eNasalance*. Future studies should explore how to optimize *mfccNasalance* by selecting the most adequate CNN model as a function of the dynamicity of the target speech data.

## Introduction

Hypernasality (HN) is one of the consequences of Velo-Pharyngeal Dysfunction (VPD). It refers to the presence of excessive nasal resonance when producing oral speech [1]. HN may be caused by anatomical malformations, as in patients with cleft palate (CP), or speech motor-

**Funding:** This research was funded by the Spanish MINISTERIO DE CIENCIA, INNOVACIÓN YUNIVERSIDADES, grant number PID2021-126366OB-I00. This funding received by Enrique Nava and Ignacio Moreno-Torres. This research was also funded by the Spanish JUNTA DE ANDALUCIA, grant number UMA18FEDERJA021. This funding received by Enrique Nava and Ignacio Moreno-Torres.

**Competing interests:** The authors have declared that no competing interests exist.

disorders [2]. Evaluating HN is most relevant for making clinical decisions and planning effective interventions, particularly in children with CP [3].

Traditionally, HN has been evaluated perceptually, with one or more expert Speech & Language Therapists (SLTs) rating the degree of nasality of the patient (e.g. on a 0–3 scale; [4]). Perceptual evaluation is still considered the gold standard for HN assessment [5]. However, perceptual evaluation is a demanding task that requires extensive training. This has motivated researchers to develop objective HN evaluation tools to support subjective evaluation. One important tool is Nasometry [6], which was developed half a century ago, but continues to be a well-considered method among SLTs.

Nasometry requires that speech is recorded using a split channel pair of microphones separated by a plate that records oral and nasal signals separately. Based on these dual-channel recordings, it is possible to compute Nasalance, which is the ratio of nose energy to mouth and nose energy in a small 300 Hz band centered around 600 Hz [5, 7]. In the remainder of this study, we refer to this measure as *eNasalance*. From a clinical perspective, Nasometry has two important characteristics that may partly explain its relative success: 1) it provides an intuitive percentage score that is easily interpretable by a clinical expert; 2) it can be used in any language or dialect, and for any utterance of interest for SLTs.

However, from today's perspective, *eNasalance* seems clearly limited, both from perceptual and technical perspectives. From a speech perception perspective, the acoustic information used to compute *eNasalance* is only a small portion of the information that the human listener has access to. For instance, vowel nasalization may impact the spectrum up to 3 kHz or more [8], a value much larger than the small band used to compute *eNasalance* [6]. Furthermore, while Nasometry is based only on the summed energy of the speech signal, humans may use varied spectral information [8]. This may partly explain the broad variance in studies that have computed the Pearson correlation between the *eNasalance* and perceptual nasality (with $r$ values ranging between 0.88 and 0.42) see [9–11]. From a technical perspective, research in the last half-century has proposed multiple speech features that have been shown to be highly effective in processing speech. One good example is Mel Frequency Cepstrum Coefficients (MFCCs), which model speech perception in humans and are commonly used in Automatic Speech Recognition tools. Furthermore, there are currently multiple machine learning (ML) algorithms that may serve to classify complex feature sets, such as MFCCs (see [12–14]), or high-speed nasopharyngoscopy [15]. Thus, it seems convenient to attempt to redefine nasalance by considering the technical and scientific advances in the last 50 years. Given that many studies in the last two decades have used advanced speech processing techniques to evaluate nasality using monophonic signals (i.e., a single speech signal), we will briefly review these models before making our own proposal.

## Previous ML proposals

The development of ML models requires a training process, during which the model is fed with the same type of information that is expected to evaluate. In the case of HN ML models, many have been trained with healthy and hypernasal utterances; in most cases, the utterances were sustained vowels (for example, [16, 17]) whereas a few studies have used monosyllabic words [18], target words [19], or a few sentences [20, 21]. This means that these models can only be used to detect/quantify nasality in these specific utterances (and hence in one language), and they lack the flexibility of Nasometry.

Other studies have adopted flexible approaches. Carignan et al. [14] uses MFCCs in combination with other acoustic features to train an ML algorithm to create nasalance-like signals and compares the results with Nasometry measures. However, they do not analyze the interest

of this approach for speech assessment, which is the main aim of this study. Siriwardena et al. [15] uses the full acoustic waveform to train an ML algorithm to model nasalance and compare the results with nasopharyngoscopy. However, in speech therapy, both Nasometry and naso-pharyngoscopy are instrumental methods, whose validity needs to be compared with the ground truth (i.e. perceptual evaluation; [1, 22]). In contrast, Mathad et al. [23] trained their model, a Deep Neural Network (DNN), exclusively with speech samples from healthy speakers, without any clinical data, and aim to measure hypernasality in children with HN pathology. The speech samples were part of a large oral corpus that had already been phonetically tran-scribed. This allowed the authors to classify speech sounds into four major groups: nasal con-sonants, nasal vowels, oral consonants, and oral vowels. Note that nasal vowels do not exist as separate phonemes in English (or in many other languages). However, because vowels in con-tact with nasal consonants tend to be nasalized (universally, although the degree varies accord-ing to language or dialect), they used these vowel fragments to create a large set of nasal vowels. After the speech data were classified into these four classes, the speech signal was seg-mented into 25 ms windows from which the 13 MFCCs and the first and second derivatives were computed. The resulting 39 coefficients served to train the DNN. The trained model was used to evaluate speech samples from healthy children and children with hypernasal speech, providing a nasality score per child. Finally, the Pearson correlation was computed between the DNN scores, and the perceptual scores produced by human experts. The authors found that the correlation was .80 with the Americleft database, a value on par with trained clinicians on this dataset.

From the perspective of the present study, Mathad et al.'s proposal has the advantage of being utterance independent. However, some aspects of this process require further consider-ation. One is that their model used a very large, annotated speech corpora, and that such cor-pora are available only for a small number of languages. In addition, the authors did not present data on alternative objective evaluation methods, such as Nasometry. Thus, it remains to clarify whether or not their model is more effective than Nasometry. Finally, from a techni-cal perspective, this model favors spectral analysis over temporal analysis. This approach is compatible with the common assumption that nasality is a spectral phenomenon, which is cer-tainly true in the case of sustained speech sounds. However, in the case of running speech, we may expect nasality characteristics (e.g., nasal formants) to exhibit important temporal varia-tion. Note also, that as standard speech evaluation protocols use different types of utterances (i.e. from sustained sounds to running speech [4]), it is possible that the type of acoustic infor-mation (e.g. spectral, temporal, or spectral-temporal) might depend on the target utterance. Thus, it seems necessary to explore whether or not the accuracy of HN ML models is influ-enced by the type of utterance. One type of ML algorithm that seems ideal for this task are Convolutional Neural Networks (CNNs), as it is possible use one specific architecture for dif-ferent types of acoustic cues (i.e. by selecting the appropriate kernel shapes). Also, CNN have been shown to successfully detect nasality, though only in mono-syllabic words [24]. These considerations led us to use CNNs in the present study.

Finally, one issue that has received little attention is whether or not HN MLs model trained with data from one dialect can be used with other dialects. This is relevant specifically for dia-lects that may vary in the degree of nasality. For instance, in the case of Spanish language, it has been observed that speakers from Central America show a relatively strong nasalization tendency, as compared with speakers from other areas (e.g. Spain, Argentina, Chile, etc. [25]). This raises the possibility speakers' subjective evaluation of HN varies between dialects, and that a ML model trained for a non-nasal dialect (e.g. Spain) might show reduced accuracy when tested with data from a nasal dialect (and vice versa). More data is needed to clarify this issue.

The main questions to be investigated in the present study are: 1) is nasalance, computed with CNNs and MFCCs (i.e. *mfccNasalance*) closer to human perception than the traditional *eNasalance*? 2) is it possible to use one model for all utterance types or, alternatively, utterance-type specific models are needed? and 3) does dialectal variation influence the accuracy of *mfccNasalance*? (i.e. is it possible to use one single model for multiple dialects or, alternatively, dialect-specific models should be trained?

## Our approach

As noted above, in this study, we adopted Convolutional Neural Network (CNN) as the ML model. CNNs were originally used for image classification [26]. However, when speech signals are converted into images (e.g., spectrograms), CNNs can be used to classify these images.

One important advantage of CNNs is that the information process is divided into two phases. In one phase, a series of convolutions detect speech features (which might be viewed as equivalent to acoustic cues in speech signals). In the second phase, the proper model is trained to classify the features identified in the convolutions. The features depend on the shape of the kernels used to convolve the images (see Section Kernels and phonetic information). Therefore, the type of kernel should be adapted to the type of information to be classified. In their model [24] concluded that the kernel producing the most optimal results was a spectral one. As noted above, this result is compatible with the common assumption that nasality is a spectral phenomenon. However, assuming that in running speech, spectral cues become dynamic [27], it is possible that temporal kernels might be more appropriate; furthermore, given that HN evaluation protocols include utterances with varying dynamicity, it is possible that the CNN results may vary depending on the utterance-type/model combination. For instance, a CNN model with spectral kernels might be appropriate for static utterances (e.g., sustained vowels and possibly short words), and one with temporal kernels might be appropriate for dynamic utterances (e.g., sentences and diadochokinetic syllables). Note that this distinction is compatible with evidence that humans have separate spectral and temporal processing skills [28].

In order to clarify the impact of dialectal variation, we explored two different train-test conditions: *same-dialect* and *different-dialect*. In the *same-dialect* condition, one dialect was used in both training and test databases. In the *different-dialect* condition, the CNN model was trained with one dialect and evaluated using a different dialect. To this end, as a part of this study, we collected three training databases respectively from Spain, Chile, and Costa Rica.

## Overview of the *mfccNasalance* model

Fig 1 shows an overview of the *mfccNasalance* model. Below, we describe the steps of the model.

## Audio input and preprocessing

The speech signal was recorded using a Nasometer (icSpeech, Rose Medical Solutions Ltd., Canterbury, UK), which produces two-channel audio signals. These signals are divided into 250 ms windows with 150 ms overlap (i.e., every window starts 100 ms after the previous one). For each window, 39 MFCCs were computed using the Librosa package in Python, with a 25 ms frame length a 10 ms overlap. This results in a three-dimension matrix: 2 (channels) × 39 MFCCs × 26 timeframes. These matrices (or two-color pictures) were input into the CNN model.

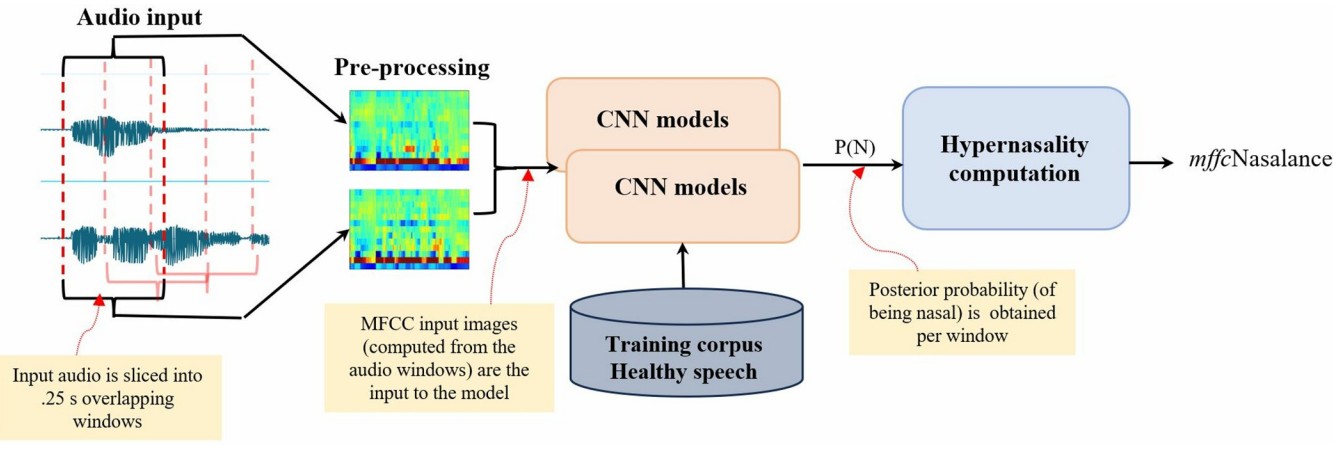

**Fig 1. Overview of the proposed CNN nasalance model for hypernasality prediction.**

## Basic architecture of the neural networks

The network model is illustrated in Fig 2. The network parameters were initialized using a standard Glorot uniform initializer. The batch size and training epochs were set to 64 and 100, respectively. The number of filters used is 16. As part of this study, we tested multiple kernels of size $i \times j$ $(1 \leq i \leq 8, 1 \leq j \leq 8)$, as explained below. Independent of the kernel shape, the stride was 1 and padding was set to *same*. A linear activation function was applied to each convolutional layer. Subsequently, batch normalization and a leaky Rectified Linear Unit were applied. In the pooling layers, a $2 \times 2$ kernel with stride 1 and the same padding was used; the method employed was max pooling. A 128-units fully connected layer with a linear activation was used. The output layer contains a softmax layer and a classification layer.

To train each model, 80% of the speakers were included in the training dataset and the remaining 20% were used to validate the model. A cross-validation process with 5 k-fold was performed, and the average of the correlation results with test data was used. The Adam optimizer was used, with a learning rate of 1e-5. The training used a single Nvidia RTX A100

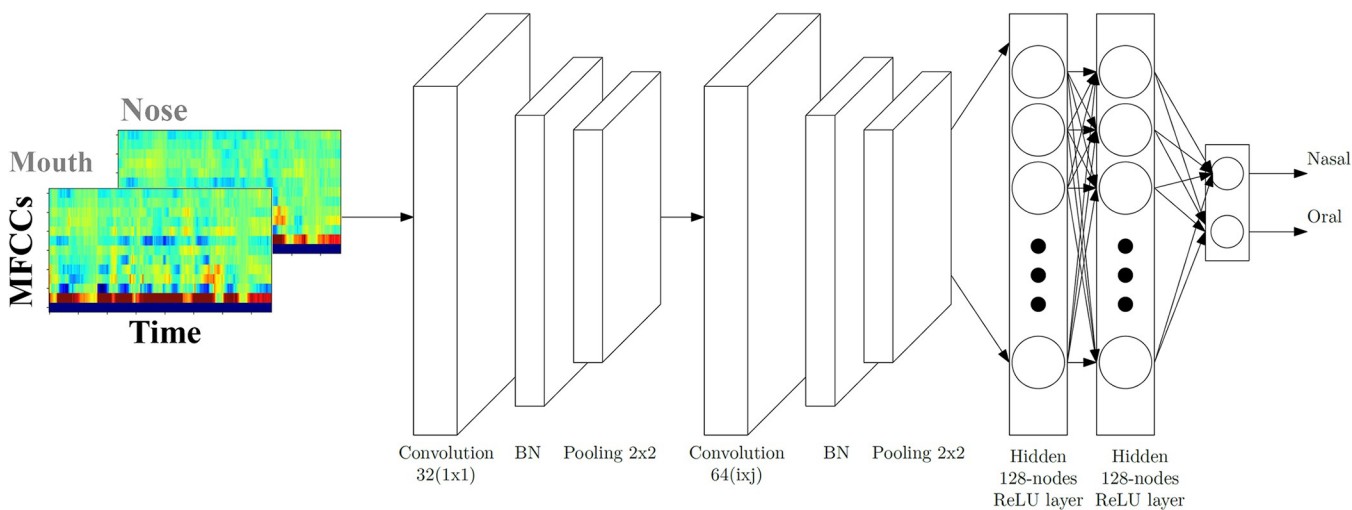

**Fig 2. Convolutional Neural Network model for hypernasality prediction.**

graphics card. The authors thankfully acknowledge the computer resources, technical expertise, and assistance provided by the SCBI (Supercomputing and Bioinformatics) Center of the University of Malaga. Script files can be found in public repository: https://github.com/Caliope-SpeechProcessingLab/Nasalance-MFCC-CNN.

## Kernels and phonetic information

As the MFCCs input images used in this study represent time in the horizontal axis and MFCCs in the vertical axis, kernel shapes can be grouped as follows (see Fig 3):

1. Spectral (cepstral): $(2 \times 1)$, $(3 \times 1)$, $(4 \times 1)$, etc.

2. Temporal: $(1 \times 2)$, $(1 \times 3)$, $(1 \times 4)$, etc.

3. Spectral-temporal: $(2 \times 2)$, $(3 \times 3)$, $(4 \times 4)$, $(2 \times 4)$, $(4 \times 2)$, etc.

Note that in the default setup, the same kernel was used for both the convolution layers. However, we decided to explore one further combination using a $1 \times 1$ kernel in the first layer, combined with any kernel shape in the second layer. This approach aims to reduce the dimensionality of the model [29, 30]. We assumed that it might serve to compare the oral and

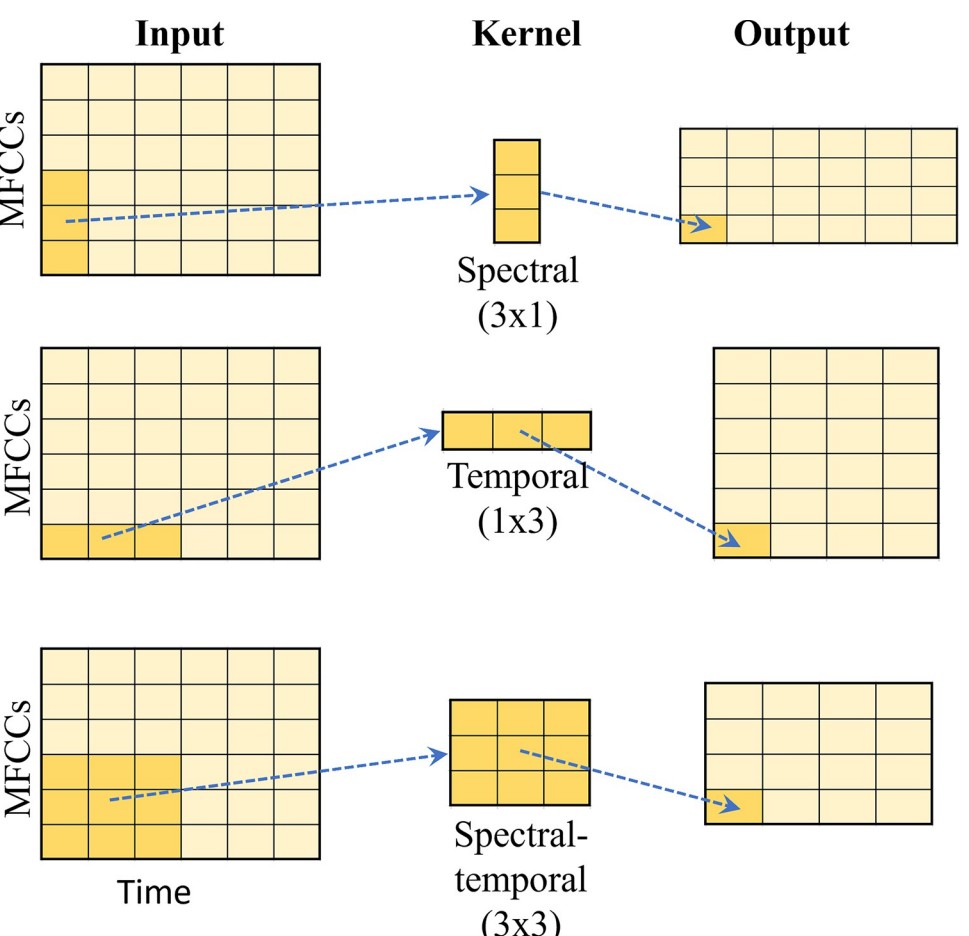

**Fig 3. Kernel shapes and phonetic information.**

nasal signals for each MFFC time point (i.e., similar to energy computation in traditional *eNasalance*). In total, we computed 126 kernel combinations; 63 of these combinations used the same kernel in both layers, and 63 more were obtained by using 1 × 1 in the first layer and any kernel size in the second layer. Of these kernels, 14 were spectral, 14 were temporal, and 98 were spectral-temporal.

## Training

The training data consisted of the MFCCs input images described above, together with the nasality value (which could be either *nasal* or *oral*) of the corresponding 250 ms window. Fig 4 shows the steps performed to determine the nasality value. First, the speech samples of the speaker were annotated using Praat v6.1.53 [31] and a phonetic transcription was obtained using the Montreal Forced Alignment v1.0.0 tool [32]. The phonetic codes were then reclassified as oral or nasal. At this stage two groups of sounds were classified as nasal: 1) the three nasal consonants (/m/, /n/ and /ɲ/); and 2) a section of the vowels that are in contact with nasal consonant (e.g. the end part of "a" in / an.tes /). The duration of this section varies with vowel duration. In this study, we assumed that the entire vowel was nasalized if it was shorter than 60 ms, 50% if it was between 60 and 90 ms, and 30% if it was longer than 90 ms. These limits are based on experiments conducted in the early design phases of this study. In the third step, the speech signal was segmented using a moving window of 250 ms length and 100 ms overlap. Next, the accumulated duration of the frames annotated as nasal was computed and divided by the total duration (i.e., 250). If the resulting value is higher than .30 (i.e. at least 75

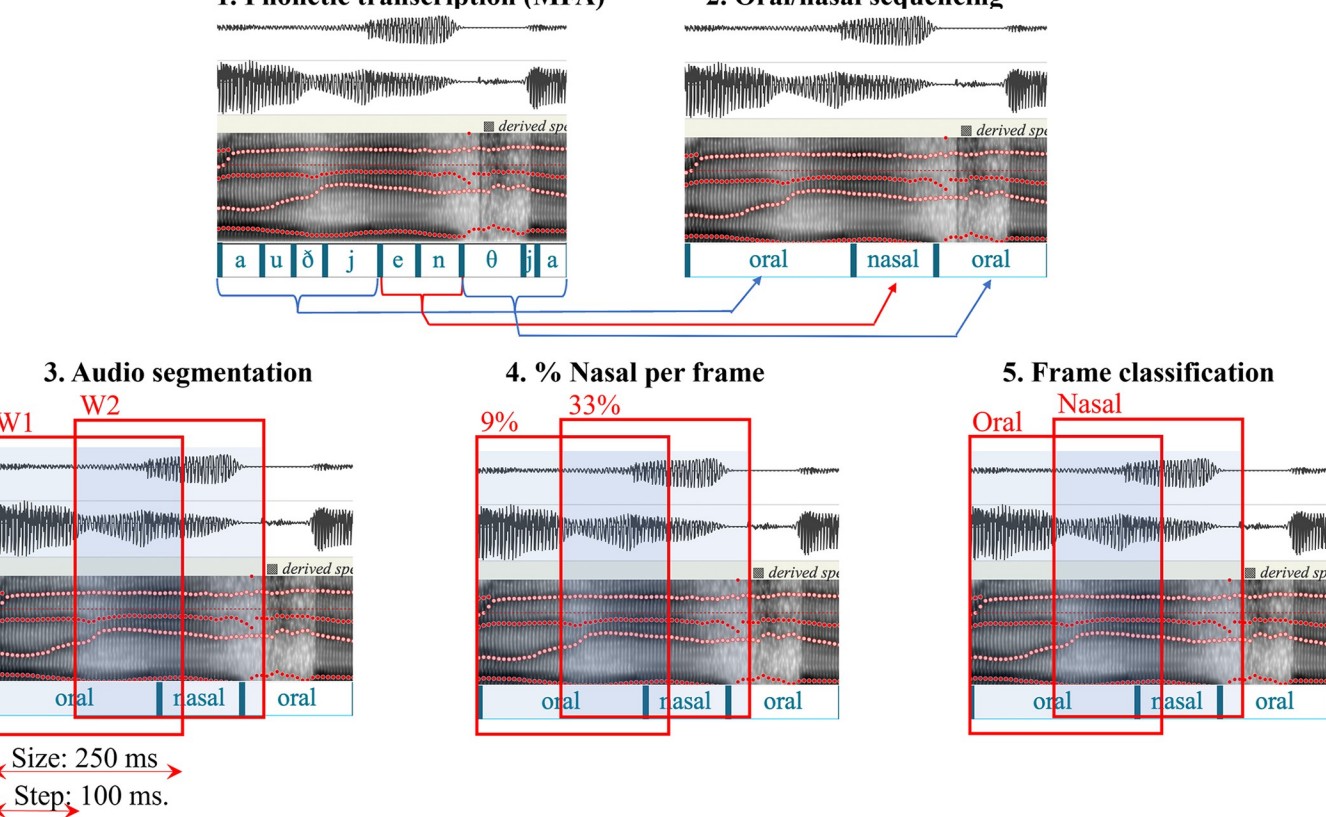

**Fig 4. Train data annotation and classification.**

ms), the window is classified as *nasal*; otherwise, it is classified as *oral*. Note that this percentage was adopted after conducting multiple experiments, suggesting that this value was optimal.

## Testing and hypernasality computation

Once the model is trained, it can be used to test new MFCC input images (from 250 ms speech samples). Testing produces the posterior probability of being nasal for each input image, with values between 0 and 1. The *mfccNasalance* for a speaker is computed by averaging the posteriors probabilities of the speech fragments of that speaker. Once the full test database has been tested (i.e. *mfccNasalance* scores have been obtained for all speakers), the speaker scores are transformed into a 0–3 scale to provide clinical meaning, like the perceptual scale. To transform the probabilities into a 4-level scale, we had to determine the boundaries between the categories. Since we already knew how many children were perceptually classified as oral or nasal, and specifically as oral, mild, moderate or severely nasal, we assumed that the same distribution would apply to the nasalance measurements and make a scale transformation maintaining the same group distribution. One limitation of this approach is that bias may have been artificially avoided. However, as we analyzed only the accuracy and the same approach was used for *eNasalance* and *mfccNasalance*, we assumed that it would be valid for this study.

## Methods

### Speech samples

The present study analyzed speech samples from a large number of healthy and hypernasal speakers. All participants, or parents/tutors in the case of minors, signed an informed consent form. The recruitment period for this study began on July 1st, 2023, and concluded on October 15th, 2023.The study was conducted in accordance with the guidelines of the Declaration of Helsinki and was approved by the Ethics Committee of the University of Málaga (protocol code 67-2023-H, June 13, 2023).

Three Spanish language speech databases were used in this study: European Spanish, Costa Rican Spanish, and Chilean Spanish. All speech samples were recorded using the Nasometer device described in Section" Audio input and preprocessing", and consisted of recordings of young, healthy women while reading four texts. Three of these four texts include a balanced representation of the inventory of Spanish language phonemes [33–35]. The fourth text was written as a part of this study and included many instances of the three nasal consonants in various contexts. A total of 50 speakers were recorded in Spain, 42 in Costa Rica, and 32 in Chile. All speakers were young adult females (age range: 18–30). The motivation to use this age range was to minimize the differences in fundamental frequency with that of the target children.

The test database included speech samples from 38 children with hypernasal speech and from 11 healthy children. The patients were recruited from Málaga and Barcelona (Spain). The children were regularly followed up by interdisciplinary teams in a clinical setting. Children from Málaga were recruited from the Materno Infantil Hospital. Children from Barcelona were recruited with the help of the local CP Association (FICAT). There were 20 males (aged 5–13 years) and 18 females (age range 4–21 years). The main criterion for inclusion was the existence of a history of HN speech. For male participants, a second criterion was having a fundamental frequency of 180 Hz or higher. Some of the patients were recorded on two occasions (N = 5) or three occasions (N = 2), for which the total number of speech samples from the HN group was 47.

The control group consisted of 11 participants with no history of speech disorders. Seven of them were male and four were female; their ages ranged from 5 to 12 years old. Individuals

**Table 1. Utterances in test data.**

| Category | Utterances |
|---|---|
| Diadochokinetic syllable repetition (+ dynamic) | *papapa…, pipipi…, tatata…, tititi…, kakaka…, kikiki…* |
| Words (-dynamic) | *boca, pie, llave, dedo, dedo, gafas, silla, sol, casa, pez* |
| Sentences (+dynamic) | *A David le duele el dedo* |
| | *Al gato de Agatha le gusta el yogur* |
| | *Uy, hay algo ahí* |
| | *Si me llevo la llave* |
| | *Susi sale sola* |
| | *Fali fue a la feria* |
| | *Los zapatos de Cecilia* |
| | *La jirafa de Jesús* |
| | *Toda tu taza de té* |
| | *Papá puede pelar a Pili* |
| | *Quique coge el papel de calco* |

with upper respiratory tract infection, mixed nasal resonance, dysphonia, hoarseness of voice, or hyponasality were excluded from the study. All participants spoke Spanish as their native language.

The test recordings were obtained using the Nasometer described above. Each participant produced 45 utterances as part of a repetition task that is routinely used in our lab to evaluate children with resonance disorders. Only a subset of these 45 utterances was used for the present study, as shown in Table 1. Note that syllable repetition includes as many repetitions of the syllable as the user is able to perform during the recording time.

## Dialect conditions

Fig 5 shows the combinations of the training and test dialects used in this study: 1) same-dialect and 2) different-dialect.

## Perceptual ratings

A three-point scale of nasality was used to rate each utterance as follows: 0 (normal), 1 (at least one instance of vowel nasalization), and 2 (at least one instance of consonant nasalization). The scores were then summed and divided by the maximum score (i.e., $2 \times$ the number of utterances). This resulted in scores ranging between 0 (maximally oral) and 1 (maximally nasal). Perceptual ratings were calculated by two SLTs who had, respectively, two and four years of experience on patients with resonance disorders. Whenever the two SLTs disagreed, a third experienced SLT determined the nasality score.

Finally, the percentage ratings were recorded using a 4-level scale. Oral: nasality $\leq 0.05$; Mild: $0.05 <$ nasality $\leq 0.25$; Moderate: $0.25 <$ nasality $\leq 0.50$; Severe: nasality $> 0.50$. Note that this rating scale, based on the expert's own clinical experience, does not differentiate between closed and open vowel nasalization, which is a common distinction in many protocols [4, 36]. However, as close vowels were included in only approximately 50% of the utterances, it was assumed that the results would reflect the close/open distinction.

### *eNasalance* ratings

Nasalance was computed using a script developed by our team using Praat software [26]. The script produced one nasalance measure per utterance following these steps: 1) the nose and

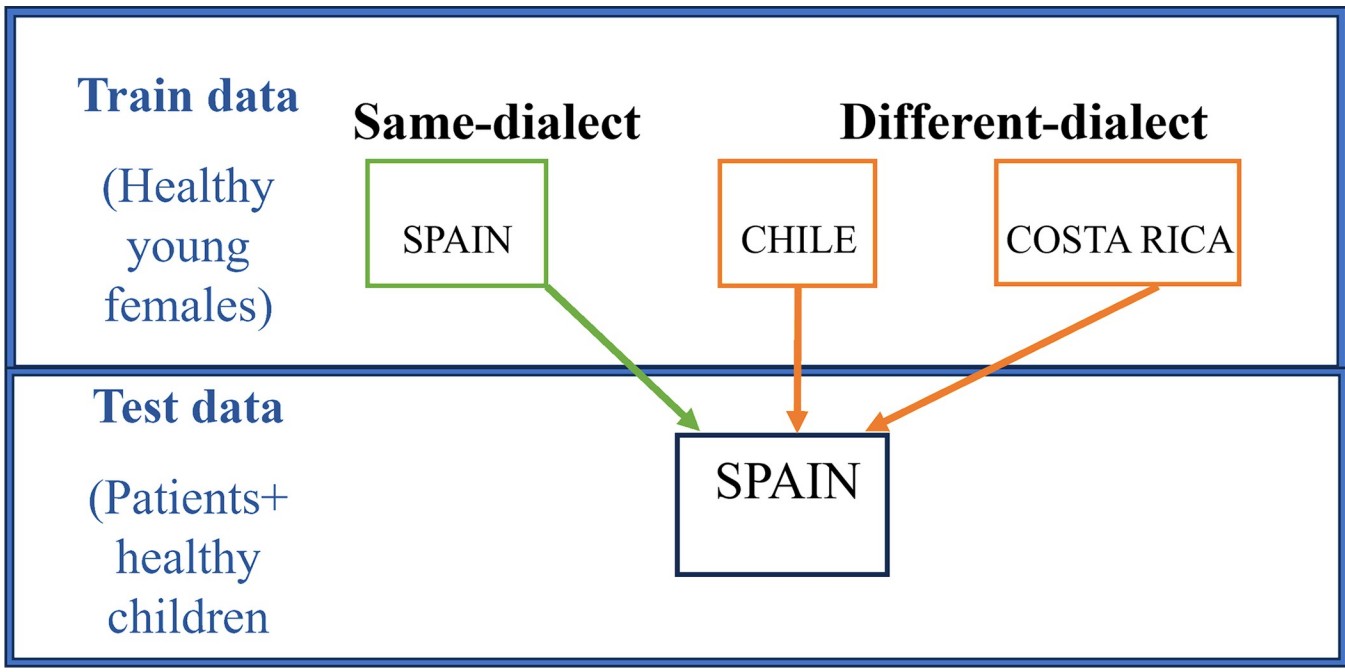

**Fig 5. Train data annotation and classification.**

mouth signals were filtered, and only a 300 Hz wide band centered at 600 Hz was retained; 2) nasalance was computed as the ratio of nasal acoustic RMS energy to the sum of oral and nasal RMS energy. For each participant, the mean nasalance value was computed by averaging the nasalance values for each utterance. The percent eNasalance scores were recoded into a 0–3 scale following the same procedure used for the mfccNasalance scores.

## Statistical analyses

Three measures were obtained for each test speaker: perceptual nasality, *eNasalance* and *mfccNasalance*. Furthermore, a new *mfccNasalance* score was obtained for each trained model and different test utterances. To compare *eNasalance* with the different *mfccNasalance* measures, we computed the Spearman correlation between these scores and perceptual ratings. Note that Pearson correlation was not used because the data did not comply with the normally distributed required condition of this statistical test. This is known by applying the Shapiro-Wilk test to the scores from syllables, words, sentences, and all together, with a p-value less than 0.05 in each case. The results for different sets of simulations (e.g., same-dialect vs different dialect) were compared using student t-test. In order to analyze the optimal CNN setup Two-way ANOVAs were run using the kernel shape (spectral, temporal, spectral-temporal) and the presence of the point-wise $1 \times 1$ kernel.

## Results

The results are organized to answer the three research questions noted above. First, we present the result for the *mfccNasalance* approach in the same-dialect condition and compare those results with *eNasalance*. Second, results are presented for multiple kernel shape × utterance type combinations (in the same-dialect condition). Finally, we analyze the results for the different-dialect condition.

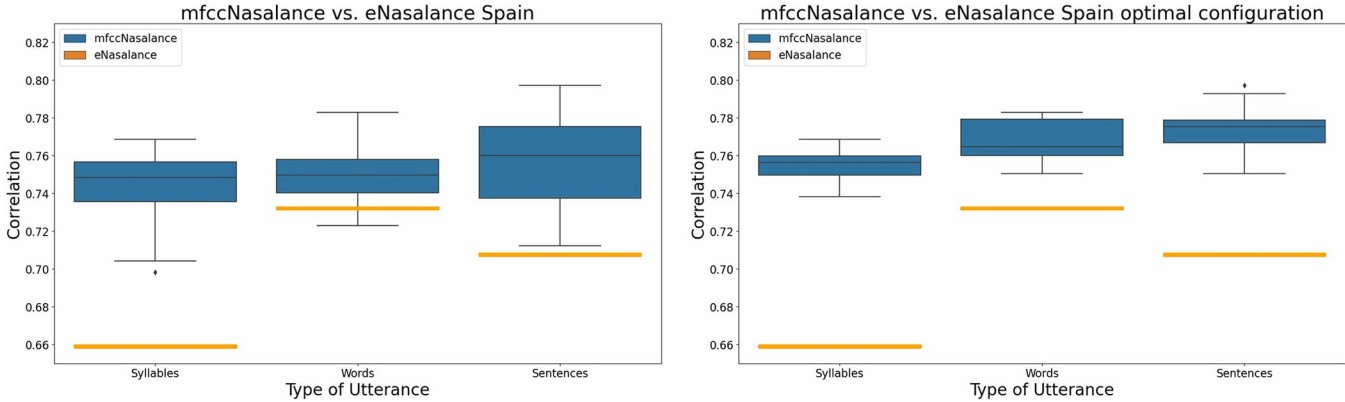

**Fig 6. Correlation between *e-Nasalance* and perceptual scores (orange rectangle), and *mfccNasalance* and perceptual scores (blue) in the same-dialect condition (Spain).** Left figure shows the results for all the CNN configurations. The right figure shows the results for the CNNs using the optimal configuration (Spain: Syllable k11 = True, Words Kernels = Temporal, Sentences k11 = True).

### *mfccNasalance* vs. *eNasalance*: Same-dialect condition

In order to compare the two measures, we trained and tested five CNN models (i.e. using a cross-correlation) per kernel combination (N = 126). Thus, the total number of models was 630. In all these cases, the train and test dataset used was European Spanish (i.e. same-dialect condition). For each of the 630 models, we computed *mfccNasalance* as well as its correlation with perceptual scores separately for the diadochokinetic syllables, words, and sentences. Finally, mean for the five scores per of each kernel combination (N = 126) × utterance type (N = 3) was obtained, which results in 378 measures (i.e. 126 per utterance type).

Fig 6 (left) shows one boxplot for each utterance type. For syllables and sentences, the 126 correlational measures for *mfccNasalance* were higher than *eNasalance*. This means that, independently of the kernel combination, *mfccNasalance* is closer to human perception than *eNasalance*. In the case of words, the advantage of *mfccNasalance* is smaller than in the case of syllables, but the scores are higher than *eNasalance* in most cases (i.e. all except the lower whisker). For sentences, *mfccNasalance* scores are better than in the case of words, though not as good as for syllables.

### Kernel combinations and utterance types in the same-dialect condition

Next, we analyzed to what extent the correlation with perceptual scores varied as a function of the kernel setup in the CNN models. Two factors were analyzed: the presence (or not) of a $1 \times 1$ kernel in the first layer (k11), and the kernel shape used in the second layer (and also in the first one if $1 \times 1$ kernel was not used). Two-way ANOVA were computed separately for syllables, words and sentences. The results showed that using a $1 \times 1$ kernel resulted in increased accuracy for syllables ($p < .000$) and sentences ($p < .000$), but nor for words. In contrast, the kernel shape had a significant impact for words ($p < .000$), but not for syllables or sentences. Post-hoc pair-wise comparisons for the word-level data showed that the difference was significant among all groups of CNN models (temporal > spectral > spectral-temporal). Fig 6 (right) shows the box-plots obtained for the optimal CNN models. The results indicate that selecting the optimal CNN setup does not have a major impact on the mean correlation; however, it reduces clearly the variability. Table 2 summarizes the results obtained in the two-way ANOVA.

**Table 2. Two-way ANOVA for Spain-trained models and optimal configuration per utterance type.**

| | Spain | | | |
|---|---|---|---|---|
| Category | k11 in first layer | Kernel shape | Interaction k11 and kernel | Optimal configuration |
| Syllable | p< .000 (post-hoc: k11) | ns | p < .01 | k11 = True |
| Words | ns | p< .000 (Temporal > Spectral > Spectral-Temporal) | ns | Kernels = Temporal |
| Sentences | p< .000 (post-hoc: k11) | ns | p < .01 | k11 = True |

ns: not significant. k11: kernel 1 × 1 in the first layer of the CNN model.

## *mfccNasalance* vs. *eNasalance*: Different-dialect condition

Next, we run the same analyses as above for the different-dialect condition (i.e. Chile and Costa Rica). First, we examined the scores for the full set of models N = 126 per dialect. The results of the Costa Rica-trained models are shown in Fig 7 (top left). Note that, compared with Spain-trained models, the results of the Costa Rica-trained models are relatively poor. For instance, for approximately half the models mfccNasalance for words was lower than eNasalance for the same words. As for Chile (Fig 7 down left), the results seem to be comparable to those of Spain. In order to confirm this apparent contrast, we computed paired T-test between, on the one hand, Spain-trained models and, on the other hand, Chile or Costa Rica-trained

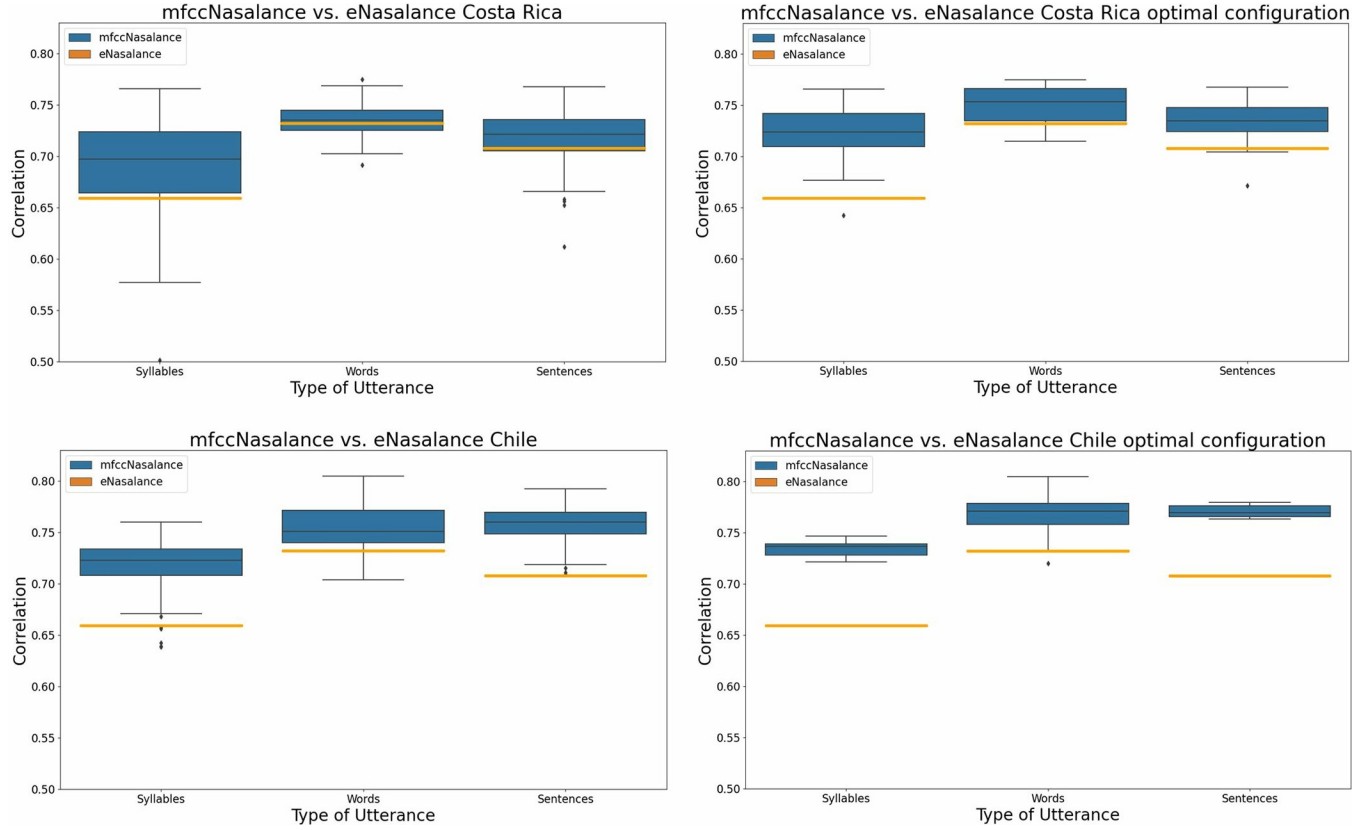

**Fig 7. Correlation between *e-Nasalance* and perceptual scores (orange rectangle), and *mfccNasalance* and perceptual scores (blue) in the different-dialect condition.** Top is for Costa Rica trained models, down for Chile. Left figures shows the results for all the CNN configurations. The right figures show the results for the CNNs using the optimal configuration (Costa Rica: Syllable k11 = True, Words Kernels = Temporal, Sentences k11 = True. Chile: Syllable Kernels = Spectral, Words k11 = False, Sentences Kernels = Spectral).

**Table 3. Two-way ANOVA for Costa Rica (top) and Chile (down) trained models.**

| Costa Rica | | | | |
|---|---|---|---|---|
| **Category** | **k11 in first layer** | **Kernel shape** | **Interaction k11 and kernel** | **Optimal configuration** |
| Syllable | p< .000 (post-hoc: k11) | ns | ns | k11 = True |
| Words | ns | p< .000 (Spectral > Spectral-Temporal > Temporal) | ns | Kernels = Temporal |
| Sentences | p< .000 (post-hoc: k11) | ns | ns | k11 = True |
| **Chile** | | | | |
| **Category** | **k11 in first layer** | **Kernel shape** | **Interaction k11 and kernel** | **Optimal configuration** |
| Syllable | ns | p< .000 (Spectral > Spectral-Temporal > Temporal) | p < .01 | Kernels = Spectral |
| Words | p< .000 (post-hoc: no k11) | ns | ns | k11 = False |
| Sentences | ns | p< .001 (Spectral > Spectral-Temporal > Temporal) | ns | Kernels = Spectral |

ns: not significant. k11: kernel 1 × 1 in the first layer of the CNN model.

models. For the pair Spain-Costa Rica, the difference was significant in all cases (p < .000) For the pair Spain-Chile, the difference was significant for syllables (p < .000) but not for words or sentences.

Next, we computed two-way ANOVAS separately for each dialect × utterance type combination. The results are summarized in Table 3. In the case of Costa Rica, using a 1 × 1 kernel had a significant and positive impact both for syllables and sentences (p < .000 in both cases), but nor for words (n.s.) In contrast, the kernel shape had a significant effect for words (p < .000), but not for syllables or sentences. Post-hoc analyses confirmed that there were significant differences among all groups (Spectral > Spectral-Temporal > Temporal). In the case of Chile, the presence of a 1 ×1 kernel produced a significant and negative effect for words (p < .000) but no effect at all for syllables or sentences. In contrast there was a main effect of the kernel shape for syllable and sentences (p < .000 in both cases) but not for words. Post-hoc analysis for the three kernel shape groups showed that differences were significant in all cases (Spectral > Spectral-Temporal > Temporal).

Finally, we compared the results of the Costa Rica (Fig 7 top right) and Chile (Fig 7 bottom right) trained optimal configuration models with the corresponding optimal Spain-trained models. As the number of models was different in dialect it was not possible to use the paired samples method; instead, we used the independent samples t-student test. The *mfccNasalance* scores obtained with European Spanish data (same-dialect) were higher than the corresponding *mfccNasalance* obtained in the different-dialect condition (both for Chile and Costa Rica). For the pair Spain-Costa Rica, this difference was significant for words and sentences (p < .000). For the pair Spain-Chile, the difference was significant only for syllables (p < .000).

## Discussion

The main results of this study are as follows: first, we found that, globally, *mfccNasalance* is closer to human perception than traditional *eNasalance*. Second, our results indicate that the different utterance types may require different CNN configurations. Third, the accuracy of *mfccNasalance* varies as a function of the dialect pair used, with results being somewhat better in the same-dialect condition than in the different-dialect condition. We discuss these results below.

### *mfccNasalance* vs *eNasalance*

The main result of this study was that the correlation between *mfccNasalance* and perceptual scores was higher than the correlation between *eNasalance* and the same perceptual scores. In

other words, *mfccNasalance* seems closer to human perception than *eNasalance*. This result was especially clear in the same dialect condition: independently of the CNN configuration *mfccNasalance* was higher than *eNasalance* (see Fig 6 left). These results are compatible with the fact that nasalance is a poor approximation to perceptual nasality, as studies comparing the nasalance scores and perceptual scores have obtained correlations ranging from non-significant to strong [37]. Nasalance computes a short spectral bandwidth [6], while Nasality processes the full spectrum [8]; furthermore, while Nasometry is based only on the summed energy in the predefined bandwidth, *mfccNasalance*, like humans, has a filterbank that can split the spectrum into a set of energy measures. Thus, it is not surprising that *mfccNasalance* is closer to human perception than *eNasalance* and, hence, that the former is a better approach to evaluate nasality than the latter.

Our proposed method has some advantages compared with that in [22]. In the first place, while Mathad et al. used a large (100 h long) database, we used relatively small datasets of speech samples (between 2h and 4h per dialect). This result is relevant because, obviously, creating one annotated speech corpora of 100 h is much more costly than creating 2-hours speech corpora. Furthermore, note that the approach used by Mathad et al. required very precise phonetic annotations (which means that it is necessary to make manual revision of the oral databases); in contrast, as we only need to know that a nasal sound was present (and not the exact location, due to the use of 250 ms speech samples), manual revision of training data can be bypassed. Altogether, this means that it might be notably easier to adapt *mfccNasalance* approach to any language or dialect than to adapt Mathad et al.'s model.

## Optimal kernel configuration and utterance types

Our second research question was if it was possible to use one model for all utterance types or, alternatively, utterance-type specific models are needed. More specifically, we speculated that as dynamic utterances (i.e. diadochokinetic syllables and sentences) are characterized by temporal variability, hypernasality would be more precisely captured by temporal kernels, while spectral kernels might be more useful for -dynamic utterances. The results in the same dialect condition revealed a contrast between the optimal models for +dynamic and–dynamic utterances, though it was not in the way we had anticipated (i.e. dynamic utterances being optimally classified with temporal kernels). Rather, the key factor was the presence or not of the 1 x 1 kernel in the first layer: in the case of dynamic utterances, 1 x 1 kernels in the first layer increased the accuracy; in the case of -dynamic utterances (i.e. words), it had no impact. As for the kernel shape in the second layer, it was relevant only in the case of words.

The $1 \times 1$ convolution kernel, also known as pointwise convolution, is generally used to increase the depth of feature maps without altering the spatial dimensions. Pointwise convolution enables the network to learn combinations of input feature maps, thus creating new feature maps that represent a blend of both the mouth and nose input channels. As shown in previous studies [29, 30] this effect is particularly useful for integrating and abstracting features across the channel dimension, leading to more robust feature representations.

Thus, the one possible interpretation for the relevance of pointwise convolution with +-dynamic utterances, but not with -dynamic ones is that in the former case the system is confronted with highly heterogeneous signals; being so heterogeneous, feature abstraction and integration may help to identify the relevant acoustic data and, hence, increase the accuracy of the system. As for words, given that variability in the signal is reduced, selecting a 1 x 1 kernel does not produce any advantage; however, the fact that temporal kernels are the most relevant suggest that nasality information is dynamically produced in short bi-syllabic words in Spanish. Note that this result (i.e. the good results with temporal kernels) do not agree with those

obtained in the only previous study that used CNN to compute nasality [18]. The authors found that spectral kernels produced the best results. The difference between the previous study and ours might be explained as follows. Nasality information is relatively weak as compared with oral information, for which the latter may partially masked the former in monophonic signals [38]. Naturally, masking cannot be complete, but it seems reasonable to assume that temporal changes (e.g. formant transitions) are more easily masked than more stable sections (e.g. center of nasal consonants). Second [18], analyzed monosyllabic words in Chinese, and these words produced in isolation may be even less dynamic than the two-syllable words used in the present study, for which they might be better classified using temporal or spectral-temporal kernels. To summarize, while in [16], the most relevant nasality information was spectral, in our case temporal information may be much more relevant.

## Dialect-effect

As for the results for utterance types and dialects, one caution must be exercised. Given the limited number of speakers per dialect, it was not possible to fully confirm whether these results reflected the properties of the set of participants for each database or the general characteristics of the dialect. However, given what we know about nasality in these dialects, a tentative explanation can be proposed for the complex pattern of the results for dialects and utterances. The most relevant result is that, independently or the kernel setup, the results in the same dialect-condition were significantly better than the results in the different-dialect conditions. Furthermore, the difference was clearer for the Spain-Costa Rica pair than for the Spain-Chile pair. This result is compatible with what we know about nasality in the three dialects. As noted above, Costa Rican Spanish is known to be highly nasal, while European and Chilean Spanish are comparatively similarly oral. Thus, it seems that the phonetic differences between these two dialects may explain the results. However, the fact that the Chile-Spain pair produced poorer results than the same dialect condition indicates that other dialectal characteristics may be relevant apart from nasality.

Another interesting result is that the optimal kernel shapes are not identical for the different-dialect condition as for the same dialect condition. Furthermore, there were differences between the two different-dialect condition. In the case of Costa Rica, the pointwise convolution has a significant positive effect for +dynamic utterances exclusively (i.e. as in the case of Spain); then, for words, spectral kernels produce the best results. In the case of Chile, the pointwise convolution produces no impact for +dynamic utterances, and a negative impact for words.

Two aspects of these results require further attention: the pointwise convolution and the preference for specific kernel shapes. As to the pointwise convolution, the results in Costa Rica are identical to those in Spain. One possible explanation for this result is that, even the two dialects might differ in nasality expression spectrally (i.e. with nasalization more pronounced in Costa Rica) but not temporally. As for Chile, the lack of significance of the pointwise convolution suggests that we find the different pattern: the two dialects (European and Chilean) might be similar in spectral terms, but not so much in temporal terms. Inasmuch as the Chilean samples are less dynamic than the European ones, the effect of the convolution might be negative in the case of -dynamic utterances, and neutral in the case of +dynamic utterances. Thus, the different-dialect data further confirms the relevance of the pointwise convolution. As to the kernel shapes, it is intriguing that temporal kernels produced the best results in the same dialect condition, and spectral kernel shapes in the different-dialect condition. Unfortunately, the fact that we found three different kernel configurations for three different dialects makes it impossible to make any generalization. To summarize, dialectal data further confirms that a

key aspect of the CNN setup is the pointwise convolution: it tends to have a significant and positive impact with highly heterogenous data, and it may also have a negative impact with less heterogeneous data (i.e. short -dynamic utterances).

## Future lines of research

The present proposal shows that *mfccNasalance* preserves the advantages of Nasometry while simultaneously incorporating more recent speech processing techniques. It also shows that in order to increase accuracy, it is convenient to use utterance and dialect specific kernel setup, and that the use of a pointwise convolution has the strongest influence on the final accuracy. These results imply that in order to develop a nasality evaluation tool that is valid for the varied utterance types used in clinical protocols (e.g. words, sentences. . .) and also for speakers from varied dialectal backgrounds, it might be necessary build more than one nasality model. Furthermore, it implies that a preprocessing stage will be needed to decide, for each target speech fragment, which is the most appropriate model. Note that this is compatible with information about speech perception by humans, as we can dynamically favor spectral or temporal speech processing [39]. Future studies should analyze to what extent it is possible to predict, for a given speech fragment, which model is the most optimal one. For this task, it seems that one alternative would be to look for inspiration from recent neurocognitive models of speech perception (e.g., [40]).

## Acknowledgments

The authors would like to express their gratitude to FICAT (Catalonia) and ASAFILAP (Andalusia).

## Author Contributions

**Conceptualization:** Enrique Nava, María Dolores García Méndez, Ignacio Moreno-Torres.

**Data curation:** Andrés Lozano, Enrique Nava, María Dolores García Méndez, Ignacio Moreno-Torres.

**Formal analysis:** Enrique Nava, Ignacio Moreno-Torres.

**Funding acquisition:** Enrique Nava, Ignacio Moreno-Torres.

**Investigation:** Andrés Lozano, Ignacio Moreno-Torres.

**Methodology:** Andrés Lozano, María Dolores García Méndez, Ignacio Moreno-Torres.

**Project administration:** Enrique Nava, Ignacio Moreno-Torres.

**Resources:** Ignacio Moreno-Torres.

**Software:** Andrés Lozano, Ignacio Moreno-Torres.

**Supervision:** Enrique Nava, María Dolores García Méndez, Ignacio Moreno-Torres.

**Validation:** María Dolores García Méndez, Ignacio Moreno-Torres.

**Visualization:** Andrés Lozano, Ignacio Moreno-Torres.

**Writing – original draft:** Andrés Lozano, Enrique Nava, María Dolores García Méndez, Ignacio Moreno-Torres.

**Writing – review & editing:** Andrés Lozano, Enrique Nava, María Dolores García Méndez, Ignacio Moreno-Torres.

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
