## [Decision Letter · Decision Letter 0]

2 Jul 2024

PONE-D-24-18061Computing nasalance with MFCCs and Convolutional Neural NetworksPLOS ONE

Dear Dr. Lozano Durán,

Thank you for submitting your manuscript to PLOS ONE. After careful consideration, we feel that it has merit but does not fully meet PLOS ONE’s publication criteria as it currently stands. Therefore, we invite you to submit a revised version of the manuscript that addresses the points raised during the review process.

This submission is interesting and timely, but several aspects of the methods and results require clarification and/or modification. Provided that the authors are willing to address them, they will be sent back to the original reviewers for re-assessment to determine suitability of the manuscript for publication.

We look forward to receiving your revised manuscript.

Kind regards,

Laura Morett

Academic Editor

PLOS ONE

2. Please note that PLOS ONE has specific guidelines on code sharing for submissions in which author-generated code underpins the findings in the manuscript. In these cases, we expect all author-generated code to be made available without restrictions upon publication of the work. Please review our guidelines at https://journals.plos.org/plosone/s/materials-and-software-sharing#loc-sharing-code and ensure that your code is shared in a way that follows best practice and facilitates reproducibility and reuse."

 [This research was funded by the Spanish MINISTERIO DE CIENCIA, INNOVACIÓN Y UNIVERSIDADES, grant number PID2021-126366OB-I00.].  

6. Please upload a copy of Supplementary files to which you refer in your text on page 12. Please amend the file type to 'Supporting Information'. If the Supplementary file is no longer to be included as part of the submission please remove all reference to it within the text.

Reviewers' comments:

Reviewer's Responses to Questions

**Comments to the Author**

1. Is the manuscript technically sound, and do the data support the conclusions?

Reviewer #1: Yes

Reviewer #2: Partly

2. Has the statistical analysis been performed appropriately and rigorously? 

Reviewer #1: No

Reviewer #2: No

3. Have the authors made all data underlying the findings in their manuscript fully available?

Reviewer #1: No

Reviewer #2: Yes

4. Is the manuscript presented in an intelligible fashion and written in standard English?

Reviewer #1: Yes

Reviewer #2: No

5. Review Comments to the Author

Reviewer #1: Questions:

"Based on these dual-channel recordings, it is possible to compute Nasalance, which is the ratio of nose energy to mouth and nose energy in a small 300 Hz band centered around 600 Hz. [5, 7]."

Q1) Why the center frequency reduced to 500Hz (in method) instead of 600Hz according to the reference [5, 7] as mentioned in the Introduction?

Q2) According to the method described by Mathad et al. (2021), MFCCs and their first and second deltas were utilized to train the DNN. In contrast, your method includes only MFCCs without the deltas, which requires further explanation. [if there is no justification provided, please implement R2]

For Revision:

R1) In comparison to the reference study by Mathad et al. 2021, which utilized a substantial 100-hour database, the training model in that study is robust enough to mitigate any subset effects. However, the current paper relies on only 2 hours of data without employing cross-validation. To enhance the reliability of the results, I recommend implementing cross-validation and conducting a straightforward comparison of accuracy (%) means to strengthen the statistical validity of the findings.

R2) The deltas of MFCCs are crucial, as demonstrated in numerous studies, for capturing the temporal dynamics that are often lost during time windowing. By including the deltas, the current spectral dimension would be expanded to a three-dimensional matrix: 2 (channels) × 39 MFCC features (13 MFCCs + 13 first deltas + 13 second deltas) × 26 timeframes. This addition would enhance the model's ability to recover temporal information.

Typos/Corrections:

T1) Ensure that all graphs include labels on the y-axis. Currently, they are missing.

T2) Address the numerous typos throughout the paper, with particular attention to consistent capitalization and formatting of "MFCCs."

Reviewer #2: The utilization of CNN on MFCC for classification and nasal and oral radiated energy based classification are well-established. So, the article has lack of novelty. The description of the optimized kernel is confusing. The author should represent the kernel with respect to individual MFCC and their derivatives. The word and phrase selection should be unified. For an example, the spectrogram and MFCC are used same purpose, which made the reader confused.

6. PLOS authors have the option to publish the peer review history of their article (what does this mean?). If published, this will include your full peer review and any attached files.

Reviewer #1: No

Reviewer #2: **Yes: **Md Mahbub Hasan

---

## [Author Response · Author response to Decision Letter 0]

16 Aug 2024

1: Manuscript have been modified to meet PLOS ONE’s style requirements..

2. Please note that PLOS ONE has specific guidelines on code sharing for submissions in which author-generated code underpins the findings in the manuscript. In these cases, we expect all author-generated code to be made available without restrictions upon publication of the work. Please review our guidelines at https://journals.plos.org/plosone/s/materials-and-software-sharing#loc-sharing-code and ensure that your code is shared in a way that follows best practice and facilitates reproducibility and reuse." 

2: All autor-generated code script files can be found in public repository: 

https://doi.org/10.6084/m9.figshare.26762818.v1

https://doi.org/10.6084/m9.figshare.26762842.v1

https://doi.org/10.6084/m9.figshare.26762833.v1

3. Thank you for stating the following financial disclosure: [This research was funded by the Spanish MINISTERIO DE CIENCIA, INNOVACIÓN Y UNIVERSIDADES, grant number PID2021-126366OB-I00.]. Please state what role the funders took in the study. If the funders had no role, please state: ""The funders had no role in study design, data collection and analysis, decision to publish, or preparation of the manuscript."" If this statement is not correct you must amend it as needed. Please include this amended Role of Funder statement in your cover letter; we will change the online submission form on your behalf. 

3: Added “The funders had no role in study design, data collection and analysis, decision to publish, or preparation of the manuscript.” in the financial disclosure.

b) If there are no restrictions, please upload the minimal anonymized data set necessary to replicate your study findings to a stable, public repository and provide us with the relevant URLs, DOIs, or accession numbers. For a list of recommended repositories, please see https://journals.plos.org/plosone/s/recommended-repositories. You also have the option of uploading the data as Supporting Information files, but we would recommend depositing data directly to a data repository if possible. 

4: Minimal anonymized data set can be found in public repository: 

https://doi.org/10.6084/m9.figshare.26762818.v1

https://doi.org/10.6084/m9.figshare.26762842.v1

https://doi.org/10.6084/m9.figshare.26762833.v1

5: Ethics statement only appear in Method section.

6. Please upload a copy of Supplementary files to which you refer in your text on page 12. Please amend the file type to 'Supporting Information'. If the Supplementary file is no longer to be included as part of the submission, please remove all reference to it within the text.

6: Supplementary files reference has been removed within the text.

Response to Reviewer 1 Comments

Comments made by the reviewer have been applied to improve the reliability of the results and the robustness of the methods applied. In particular, the use of 39MFCC, the cross-validation of the data, and an improvement in the depth of the statistical analysis have led to a substantial change in the results obtained.

Therefore, in this revised version of the article, extensive changes have been made in the Results and Discussion sections, modifying the images generated and the way of showing and justifying the results obtained.

Questions

Question 1: Why the center frequency reduced to 500Hz (in method) instead of 600Hz according to the reference [5, 7] as mentioned in the Introduction?

Response 1: Changed to: “600 Hz” in Methods.

Question 2: According to the method described by Mathad et al. (2021), MFCCs and their first and second deltas were utilized to train the DNN. In contrast, your method includes only MFCCs without the deltas, which requires further explanation. [if there is no justification provided, please implement R2]

Response 2: In the present version, training is run using 39 MFCCs coefficient with delta and second-delta in the method section, as suggested by both reviewers.

Revisions

Revision 1: In comparison to the reference study by Mathad et al. 2021, which utilized a substantial 100-hour database, the training model in that study is robust enough to mitigate any subset effects. However, the current paper relies on only 2 hours of data without employing cross-validation. To enhance the reliability of the results, I recommend implementing cross-validation and conducting a straightforward comparison of accuracy (%) means to strengthen the statistical validity of the findings.

Response 1: Cross-validation process has been implemented for train and validation speakers.

Revision 2: The deltas of MFCCs are crucial, as demonstrated in numerous studies, for capturing the temporal dynamics that are often lost during time windowing. By including the deltas, the current spectral dimension would be expanded to a three-dimensional matrix: 2 (channels) × 39 MFCC features (13 MFCCs + 13 first deltas + 13 second deltas) × 26 timeframes. This addition would enhance the model's ability to recover temporal information.

Response 2: Thanks for your suggestion. 39 MFCCs coefficient with delta and second-delta are computed to generate a 2-channel image 2 (channels) × 39 MFCC × 26 timeframes as input data of CNN.

Typos

Typos 1: Ensure that all graphs include labels on the y-axis. Currently, they are missing. 

Response 1: Graphs has been modified to include y-axis labels.

Typos 2: Address the numerous typos throughout the paper, with particular attention to consistent capitalization and formatting of "MFCCs."

Response 1: Typos corrected to use consistent “MFCCs”

Response to Reviewer 2 Comments

Questions

Question 1: The utilization of CNN on MFCC for classification and nasal and oral radiated energy-based classification are well-established. So, the article has lack of novelty. The description of the optimized kernel is confusing. The author should represent the kernel with respect to individual MFCC and their derivatives. The word and phrase selection should be unified. For an example, the spectrogram and MFCC are used same purpose, which made the reader confused.

Response 1: The description of the optimized kernel has been modified in the Results section.

MFCCs features with delta and second delta has been computed and added as input of CNN.

Word and phrase selection remains unchanged because the results show different behaviour in terms of the optimal kernel configuration for each type of utterance, which we consider worthy of distinction and discussion.

Spectrogram and MFCC are no longer used for the same purpose. Input data generated with MFCCs coefficients are defined as ‘MFCCs input images’ in the text.

---

## [Decision Letter · Decision Letter 1]

8 Oct 2024

PONE-D-24-18061R1Computing nasalance with MFCCs and Convolutional Neural NetworksPLOS ONE

Dear Dr. Lozano Durán,

Thank you for submitting your manuscript to PLOS ONE. After careful consideration, we feel that it has merit but does not fully meet PLOS ONE’s publication criteria as it currently stands. Therefore, we invite you to submit a revised version of the manuscript that addresses the points raised during the review process.

**Because neither of the original reviewers were able to re-review this manuscript, a third reviewer was recruited. This reviewer raises some important points that should be addressed, including coverage of relevant literature, data coding and analysis, and interpretation of results.  Please revise the manuscript to address the points raised by R3 and it will be reassessed for suitability for publication in PLOS One.**

We look forward to receiving your revised manuscript.

Kind regards,

Laura Morett

Academic Editor

PLOS ONE

Reviewers' comments:

Reviewer's Responses to Questions

**Comments to the Author**

1. If the authors have adequately addressed your comments raised in a previous round of review and you feel that this manuscript is now acceptable for publication, you may indicate that here to bypass the “Comments to the Author” section, enter your conflict of interest statement in the “Confidential to Editor” section, and submit your "Accept" recommendation.

Reviewer #3: (No Response)

2. Is the manuscript technically sound, and do the data support the conclusions?

Reviewer #3: Partly

3. Has the statistical analysis been performed appropriately and rigorously? 

Reviewer #3: Yes

4. Have the authors made all data underlying the findings in their manuscript fully available?

Reviewer #3: Yes

5. Is the manuscript presented in an intelligible fashion and written in standard English?

Reviewer #3: Yes

6. Review Comments to the Author

**Reviewer #3:** This study attempt to create a nasalance-like metric directly from acoustics, using deep learning models. The study is quite nice in many respects, but there are issues that would need to be addressed before publication can be considered. These issues are listed below in the order (broadly) that they appear in the paper.

Abstract, Table 1, elsewhere: "The test data were recorded in Spain and included short words (-dynamic), sentences (+dynamic), and diakinetic syllables (+dynamic)"

-What do these +/- labels mean? Why is the /ka/ in "ka" considered "dynamic" but the /ka/ in "boka" isn't? These terms are never defined or justified.

-I am not familiar with the term "diakinetic". Do you mean "diadochokinetic"?

p.3: "To our knowledge, only one ML model has the flexibility of Nasometry [18]"

-The authors may want to be aware of similar works that have previous been carried out, for example Siriwardena et al. (2024) and Carignan (2021), the latter of which has been extended to low-resource languages (Carignan et al., 2023) and hyper-nasality in children (Fagniart et al., 2024), two goals which the current study also aims to meet.

Carignan, C. (2021). "A practical method of estimating the time-varying degree of vowel nasalization from acoustic features", J. Acoust. Soc. Am. 149 (2), 911–922. doi: https://doi.org/10.1121/10.0002925

Carignan, C., Chen, J., Harvey, M., Stockigt, C., Simpson, J. & Strangways, S. (2023). "An investigation of the dynamics of vowel nasalization in Arabana using machine learning of acoustic features”, Laboratory Phonology 14(1). doi: https://doi.org/10.16995/labphon.9152

Fagniart, S., Charlier, B., Delvaux, V., Huberlant, A., Harmegnies, B.G., Piccaluga, M., & Huet, K. (2024). "Consonant and vowel production in children with cochlear implants: acoustic measures and multiple factor analysis", Front. Audiol. Otol. 2:1425959. doi: 10.3389/fauot.2024.1425959

Siriwardena, Y.M., Boyce, S.E., Tiede, M.K., Oren, L., Fletcher, B., Stern, M., & Espy-Wilson, C.Y. (2024). "Speaker-independent speech inversion for recovery of velopharyngeal port constriction degree", J. Acoust. Soc. Am. 156 (2), 1380–1390. doi: https://doi.org/10.1121/10.0028124

-You say that European Spanish is "a non-nasal dialect" (p.3) and that "Costa Rican Spanish is known to be highly nasal, while European and Chilean Spanish are comparatively similarly oral" (p.15), but how do these statements align with the statement that "vowels in contact with nasal consonants tend to be nasalized (universally)" (p.2)?

p.4: "These signals are divided into 250 ms windows with 100 ms overlap (i.e., every window starts 100 ms after the previous one)"

-Based on this description, it would seem that every window starts 150 ms (not 100 ms) after the previous one.

p.6: "(/m/, /n/ and /ñ/)"

-"/ñ/" is not a symbol used in the IPA to denote any known consonant

p.6: "In this study, we assumed that the entire vowel was nasalized if it was shorter than 60 ms, 50% if it was between 60 and 90 ms, and 30% if it was longer than 90 ms."

-Based on what evidence?

p.7: "The transformation from a percentage to a 4-level scale required deciding the limits among classes. As we knew in advance how many children were classified perceptually as oral or nasal, or as Oral, Mild, Moderate or Severely nasal, we assumed that the same numbers would apply in the case of nasalance measures. Thus, after every simulation, the results were sorted per mfccNasalance, with the first N cases being categorized as Healthy, the next M cases as Mild, etc."

-This should be explained much more clearly, I don't understand this at all. Does this just meant that you assigned identical labels as the perceptual labels?

p.9: "Fiinally, the percentage ratings were recoded using a 4-level scale. Oral: nasality ≤ 0.05; Mild: 0.05 < nasality ≤ 0.25; Moderate: 0.25 < nasality ≤ 0.50; Severe: nasality > 0.50."

-Based on what evidence?

p.9: "Nasalance was computed as the ratio of nasal acoustic energy to the sum of oral and nasal energy"

-Please specify that you used RMS energy, as shown in the Praat script.

Fig.6, etc.: Why not two sets of box plots, or two sets of violin plots, etc.? Why one box plot and one line?

Fig.6, etc.: The caption says "Correlation between e-Nasalance [...] and mfccNasalance", but this is not accurate. They are separate correlations between e-Nasalance and perceptual scores, and mfccNasalance and perceptual scores.

p.10: "the scores are higher than eNasalance in most cases (i.e. all except the lower 25th percentile)."

-Are you referring to the lower whisker? The whisker is not the 25th percentile.

Table 2: Do you mean "p" instead of "s"? Also, what is "k11"?

p.13: "(Fig. 7 down right)" -> "bottom right"

p.14: "These results are compatible with the fact that nasalance is a poor approximation to perceptual nasality"

-Citation needed. Nasalance arose as a methodology precisely because it approximates perceptual nasality.

p.14: "in contrast, as we only need to know that a nasal sound was present (and not the exact location)"

-Maybe I've misunderstood, but this doesn't seem correct, because you based your labels off of phone-wise segmentation. In order to have phone-wise segmentation, you need to know the exact location of that phone in the speech stream.

-Punctuation errors throughout.

7. PLOS authors have the option to publish the peer review history of their article (what does this mean?). If published, this will include your full peer review and any attached files.

Reviewer #3: No

---

## [Author Response · Author response to Decision Letter 1]

17 Oct 2024

Response to Reviewer 3 Comments

Comments made by the reviewer have been applied to correct erroneous terminology used, complement the referenced bibliography, and clarify parts of the paper for better understanding.

Questions

Question 1: Abstract, Table 1, elsewhere: "The test data were recorded in Spain and included short words (-dynamic), sentences (+dynamic), and diakinetic syllables (+dynamic)"

-What do these +/- labels mean? 

- Why is the /ka/ in "ka" considered "dynamic" but the /ka/ in "boka" isn't? These terms are never defined or justified.

- I am not familiar with the term "diakinetic". Do you mean "diadochokinetic"?

Response 1: Label means: more (+) and less (-). Abstract is modified to include clarification.

Utterance in Table 1, Diakinetic syllable repetition is modified. Instead of /ka/ is marked as /kakaka…/, and a paragraph is added to clarify that it represents many repetitions of syllable /ka.

The term diakinetic is replaced by diadochokinetic.

Question 2: p.3: "To our knowledge, only one ML model has the flexibility of Nasometry [18]"

-The authors may want to be aware of similar works that have previous been carried out, for example Siriwardena et al. (2024) and Carignan (2021), the latter of which has been extended to low-resource languages (Carignan et al., 2023) and hyper-nasality in children (Fagniart et al.,2024), two goals which the current study also aims to meet.

Carignan, C. (2021). "A practical method of estimating the time-varying degree of vowel nasalization from acoustic features", J. Acoust. Soc. Am. 149 (2), 911–922. doi: https://doi.org/10.1121/10.0002925

Carignan, C., Chen, J., Harvey, M., Stockigt, C., Simpson, J. & Strangways, S. (2023). "An investigation of the dynamics of vowel nasalization in Arabana using machine learning of acoustic features”, Laboratory Phonology 14(1). doi: https://doi.org/10.16995/labphon.9152

Fagniart, S., Charlier, B., Delvaux, V., Huberlant, A., Harmegnies, B.G., Piccaluga, M., & Huet,K. (2024). "Consonant and vowel production in children with cochlear implants: acoustic measures and multiple factor analysis", Front. Audiol. Otol. 2:1425959. doi:10.3389/fauot.2024.1425959

Siriwardena, Y.M., Boyce, S.E., Tiede, M.K., Oren, L., Fletcher, B., Stern, M., & Espy-Wilson,C.Y. (2024). "Speaker-independent speech inversion for recovery of velopharyngeal port constriction degree", J. Acoust. Soc. Am. 156 (2), 1380–1390. doi: https://doi.org/10.1121/10.0028124

Response 2: These related works have been added in the Introduction section.

Question 3: You say that European Spanish is "a non-nasal dialect" (p.3) and that "Costa Rican Spanish is known to be highly nasal, while European and Chilean Spanish are comparatively similarly oral"(p.15), but how do these statements align with the statement that "vowels in contact with nasal consonants tend to be nasalized (universally)" (p.2)?

Response 3: The fact that vowels in contact with a nasal tend to nasalise is universal, although dialects and languages vary in the degree to which this occurs, as is in the case of European and Chilean Spanish.

Changes in text are added: “However, because vowels in contact with nasal consonants tend to be nasalized (universally, although the degree varies according to language or dialect), they used these vowel fragments to create a large set of nasal vowels”.

Question 4: p.4: "These signals are divided into 250 ms windows with 100 ms overlap (i.e., every window starts 100 ms after the previous one)"-Based on this description, it would seem that every window starts 150 ms (not 100 ms) after the previous one. 

Response 4: Every window starts 100 ms after the previous one. Text modified to be accurate to 150 ms overlap.

Question 5: p.6: "(/m/, /n/ and /ñ/)"

-"/ñ/" is not a symbol used in the IPA to denote any known consonant.

Response 5: /ñ/ is replaced by the IPA symbol /ɲ/ .

Question 6: p.6: "In this study, we assumed that the entire vowel was nasalized if it was shorter than 60 ms, 50% if it was between 60 and 90 ms, and 30% if it was longer than 90 ms."

-Based on what evidence?

Response 6: This distribution of vowel nasalisation length is based on experiments conducted in the early design phases of the present study. A subset of the training group is selected and trained with an automatic classification system, the aim being to select different nasalised vowel lengths so that the output corresponds to the annotation produced by the speech specialist. The results shown in the paper are the ones that obtain an optimal result.

Text is modified to add clarification.

Question 7: p.7: "The transformation from a percentage to a 4-level scale required deciding the limits among classes. As we knew in advance how many children were classified perceptually as oral or nasal, or as Oral, Mild, Moderate or Severely nasal, we assumed that the same numbers would apply in the case of nasalance measures. Thus, after every simulation, the results were sorted per mfccNasalance, with the first N cases being categorized as Healthy, the next M cases as Mild, etc."

-This should be explained much more clearly, I don't understand this at all. Does this just meant that you assigned identical labels as the perceptual labels? 

Response 7: The CNN output produces values between 0 and 1, which have no clinical value. Therefore, the output is transformed into a 4-value scale. This transformation is done keeping the original training case distribution, from which we know the classification as oral, mild, moderate or severely nasal.

Text has been modified to clarify it.

Question 8: p.9: "Finally, the percentage ratings were recoded using a 4-level scale. Oral: nasality ≤ 0.05;Mild: 0.05 < nasality ≤ 0.25; Moderate: 0.25 < nasality ≤ 0.50; Severe: nasality > 0.50."

-Based on what evidence?

Response 8: This measure is supported by clinical practice experience in our lab. Experience shows that: 

- Any healthy speaker can nasalize occasionally (hence 5% seems a reasonable limit).

- Nasalizing over 50% of the speech utterances is evidence of severe hypernasality.

- Intervals between 5% and 50% are divided equally.

A clarification is added in this section.

Question 9: p.9: "Nasalance was computed as the ratio of nasal acoustic energy to the sum of oral and nasal energy”.

-Please specify that you used RMS energy, as shown in the Praat script.

Response 9: RMS energy use is specified.

Question 10: Fig.6, etc.: Why not two sets of box plots, or two sets of violin plots, etc.? Why one box plot and one line? 

Response 10: The line represents the eNasalance correlation with the perceptual nasality scores of human experts, it only has one measure, that’s why is represented as a line.

The boxplots represent the mfccNasalance correlation with human experts. They have 126 results per utterance type, as we made a k-fold = 5 cross-correlation with 126 kernel combinations and compute the mean value for kernel.

The main purpose of Figure 6 was to illustrate the correlation between eNasalance and human experts, and mfccNasalance with human experts. Separate images are created to illustrate the cases with all available data (left), while the figure on the right shows the cases with the optimal configuration: Spain: Syllable k11 = True, Words Kernels = Temporal, Sentences k11 = True.

Question 11: Fig.6, etc.: The caption says, "Correlation between e-Nasalance [...] and mfccNasalance", but this is not accurate. They are separate correlations between e-Nasalance and perceptual scores, and mfccNasalance and perceptual scores. 

Response 11: Fig. 6 and 7 captions are modified to include: “Correlation between e-Nasalance and perceptual scores (orange rectangle), and mfccNasalance and perceptual scores (blue)”

Question 12: p.10: "the scores are higher than eNasalance in most cases (i.e. all except the lower 25thpercentile)."

-Are you referring to the lower whisker? The whisker is not the 25th percentile.

Response 12: Sentence is modified to indicate: “i.e. all except the lower whisker”.

Question 13: Table 2: Do you mean "p" instead of "s"? Also, what is "k11"? 

Response 13: “s” is changed to “p”. 

k11 denoted the presence of a 1 × 1 kernel in the first layer of the CNN model. Clarification added in Table 2 and Table 3.

Question 14: p.13: "(Fig. 7 down right)" -> "bottom right" 

Response 14: Text is corrected.

Question 15: p.14: "These results are compatible with the fact that nasalance is a poor approximation to perceptual nasality"

-Citation needed. Nasalance arose as a methodology precisely because it approximates perceptual nasality.

Response 15: Citation is added. Studies comparing the nasalance scores and perceptual scores have obtained correlations ranging from non-significant to strong. 

Liu, Y.; Lee, S.A.S.; Chen,W. The correlation between perceptual ratings and nasalance scores in resonance disorders: A systematic review. J. Speech Lang. Hear. Res. 2022, 65, 2215–2234. [CrossRef] [PubMed]

Question 16: p.14: "in contrast, as we only need to know that a nasal sound was present (and not the exact location)"

-Maybe I've misunderstood, but this doesn't seem correct, because you based your labels off of phone-wise segmentation. In order to have phone-wise segmentation, you need to know the exact location of that phone in the speech stream.

Response 16: In contrast to Mathad et al., who use a 25 ms window, the present work uses a CNN with a larger window, 250 ms, which allows for less accurate annotation. 

In the present work the classification of a sound as oral/nasal is based on the percentage of nasality present, so an error in the precise location of the nasal sound does not prevent the 250ms fragment from being classified correctly.

A clarification is added in this section.

Question 17: Punctuation errors throughout. 

Response 17: Punctuation errors are corrected.

---

## [Decision Letter · Decision Letter 2]

6 Nov 2024

PONE-D-24-18061R2Computing nasalance with MFCCs and Convolutional Neural NetworksPLOS ONE

Dear Dr. Lozano Durán,

Thank you for submitting your manuscript to PLOS ONE. After careful consideration, we feel that it has merit but does not fully meet PLOS ONE’s publication criteria as it currently stands. Therefore, we invite you to submit a revised version of the manuscript that addresses the points raised during the review process.

We look forward to receiving your revised manuscript.

Kind regards,

Laura Morett

Academic Editor

PLOS ONE

Journal Requirements:

**Additional Editor Comments:**

I thank the authors for their attention to the points raised in the last round of reviews. I agree with R3 that the revisions largely address the points raised by the reviewers, and I ask that they address the remaining points raised by R3. Provided they do so, I will render a decision without re-sending the manuscript for review.

Reviewers' comments:

Reviewer's Responses to Questions

**Comments to the Author**

1. If the authors have adequately addressed your comments raised in a previous round of review and you feel that this manuscript is now acceptable for publication, you may indicate that here to bypass the “Comments to the Author” section, enter your conflict of interest statement in the “Confidential to Editor” section, and submit your "Accept" recommendation.

Reviewer #3: (No Response)

2. Is the manuscript technically sound, and do the data support the conclusions?

Reviewer #3: Yes

3. Has the statistical analysis been performed appropriately and rigorously? 

Reviewer #3: Yes

4. Have the authors made all data underlying the findings in their manuscript fully available?

Reviewer #3: Yes

5. Is the manuscript presented in an intelligible fashion and written in standard English?

Reviewer #3: Yes

6. Review Comments to the Author

Reviewer #3: Some minor issues still remain from my previous comments, but by and large the authors have done an admirable job with the revision!

p.2: "Furthermore, there are currently multiple machine learning (ML) algorithms that may serve to classify complex feature sets, such as MFCCs (see [12-14]), or high-speed nasopharyngoscopy [15]."

-The authors seem to have missed the point in providing these additional references, which are not simply for "classify[ing] complex feature sets". Carignan (2021) uses MFCCs in combination with other acoustic features to train an ML algorithm to create nasalance-like signals, and compares the results with nasometry measures. Siriwardena et al. (2024) use the full acoustic waveform to train an ML algorithm to create nasalance-like signals, and compare the results with nasopharyngoscopy. Therefore, both of these works have previously used speech acoustics to train ML algorithms to create nasalance-like signals, in a similar manner to Mathad et al. (2021) and to the current study, which is somewhat at odds with the claim that "only one ML model has the flexibility of Nasometry" (p. 2); both Carignan (2021) and Siriwardena et al. (2024) were carried out very specifically to generate nasalance-like signals from speech acoustics using ML algorithms.

Response 13: “s” is changed to “p”

-"s" still remains in the text at many different points

7. PLOS authors have the option to publish the peer review history of their article (what does this mean?). If published, this will include your full peer review and any attached files.

Reviewer #3: No

---

## [Author Response · Author response to Decision Letter 2]

8 Nov 2024

Response to Reviewer 3 Comments

New comments made by the reviewer have been applied to correct issues that remain from previous review. Minor typos are corrected, and additional text has been added.

Questions

Question 1: p.2: "Furthermore, there are currently multiple machine learning (ML) algorithms that may serve to classify complex feature sets, such as MFCCs (see [12-14]), or high-speed nasopharyngoscopy [15]."

-The authors seem to have missed the point in providing these additional references, which are not simply for "classify[ing] complex feature sets". 

Carignan (2021) uses MFCCs in combination with other acoustic features to train an ML algorithm to create nasalance-like signals and compares the results with nasometry measures. 

Siriwardena et al. (2024) uses the full acoustic waveform to train an ML algorithm to create nasalance-like signals and compare the results with nasopharyngoscopy. 

Therefore, both of these works have previously used speech acoustics to train ML algorithms to create nasalance-like signals, in a similar manner to Mathad et al. (2021) and to the current study, which is somewhat at odds with the claim that "only one ML model has the flexibility of Nasometry" (p. 2); both Carignan (2021) and Siriwardena et al.(2024) were carried out very specifically to generate nasalance-like signals from speech acoustics using ML algorithms.

Response 1: Thank you for the clarification, text has been adapted to the suggestions mentioned above.

Text added in section “Previous ML proposals”:

Other studies have adopted flexible approaches. Carignan et al. [14] uses MFCCs in combination with other acoustic features to train an ML algorithm to create nasalance-like signals and compares the results with Nasometry measures. However, they do not analyse the interest of this approach for speech assessment, which is the main aim of this study. Siriwardena et al. [15] uses the full acoustic waveform to train an ML algorithm to model nasalance and compare the results with nasopharyngoscopy. However, in speech therapy, both Nasometry and nasopharyngoscopy are instrumental methods, whose validity needs to be compared with the ground truth (i.e. perceptual evaluation; [1, 23]). In contrast to all other models, Mathad et al. [22] trained their model, a Deep Neural Network (DNN), exclusively with speech samples from healthy speakers, without any clinical data, and aim to measure hypernasality in children with HN pathology.

Question 2: Response 13: “s” is changed to “p”

-"s" remains in the text at many different points

Response 2: The remaining instances of “s” have been changed to “p”.

---

## [Editor Report · Decision Letter 3]

26 Nov 2024

Computing nasalance with MFCCs and Convolutional Neural Networks

PONE-D-24-18061R3

Dear Dr. Lozano Durán,

We’re pleased to inform you that your manuscript has been judged scientifically suitable for publication and will be formally accepted for publication once it meets all outstanding technical requirements.

Kind regards,

Laura Morett

Academic Editor

PLOS ONE

Additional Editor Comments (optional):

I thank the authors for their revisions in response to the remaining points raised by R3. The manuscript can now be accepted for publication in PLOS One.
---

## [Editor Report · Acceptance letter]

6 Dec 2024

PONE-D-24-18061R3 

PLOS ONE

Dear Dr. Lozano Durán, 

I'm pleased to inform you that your manuscript has been deemed suitable for publication in PLOS ONE. Congratulations! Your manuscript is now being handed over to our production team.

Kind regards, 

on behalf of

Dr. Laura Morett 

Academic Editor

PLOS ONE